# Characterization of Adenovirus 5 E1A Exon 1 Deletion Mutants in the Viral Replicative Cycle

**DOI:** 10.3390/v12020213

**Published:** 2020-02-14

**Authors:** Rita Costa, Nikolas Akkerman, Drayson Graves, Leandro Crisostomo, Scott Bachus, Peter Pelka

**Affiliations:** 1Department of Microbiology, University of Manitoba, 45 Chancellor’s Circle, Buller Building Room 427, Winnipeg, MB R3T 2N2, Canada; costar@myumanitoba.ca (R.C.); umakkern@myumanitoba.ca (N.A.); gravesd@myumanitoba.ca (D.G.); umcrisol@myumanitoba.ca (L.C.); bachuss@myumanitoba.ca (S.B.); 2Department of Medical Microbiology, University of Manitoba, Winnipeg, MB R3T 2N2, Canada

**Keywords:** adenovirus, E1A, exon 1, replication

## Abstract

Human adenovirus infection is driven by Early region 1A (E1A) proteins, which are the first proteins expressed following the delivery of the viral genome to the cellular nucleus. E1A is responsible for reprogramming the infected cell to support virus replication alongside the activation of expression of all viral transcriptional units during the course of the infection. Although E1A has been extensively studied, most of these studies have focused on understanding the conserved region functions outside of a full infection. Here, we investigated the effects of small deletions in E1A exon 1 on the viral replicative cycle. Almost all deletions were found to have a negative impact on viral replication with the exception of one deletion found in the mutant *dl*1106, which replicated better than the wild-type E1A expressing *dl*309. In addition to growth, we assessed the virus mutants for genome replication, induction of the cytopathic effect, gene and protein expression, sub-cellular localization of E1A mutant proteins, induction of cellular S-phase, and activation of S-phase specific cellular genes. Importantly, our study found that virus replication is likely limited by host-specific factors, rather than specific viral aspects such as the ability to replicate genomes or express late proteins, after a certain level of these has been expressed. Furthermore, we show that mutants outside of the conserved regions have significant influence on viral fitness. Overall, our study is the first comprehensive evaluation of the *dl*1100 series of exon 1 E1A deletion mutants in viral fitness and provides important insights into the contribution that E1A makes to viral replication in normal human cells.

## 1. Introduction

Adenovirus is a double stranded DNA virus that heavily relies on the host to facilitate and enable viral replication [1]. Following the delivery of the human adenovirus (HAdV) genome to the nucleus, on average approximately two hours after infection but with substantial heterogeneity to the timing [2,3], expression of viral genes will commence. The first transcripts made during HAdV5 infection are the non-coding RNAs, Virus Associated RNA I and II, occurring within the first four hours after infection [4], followed by the expression of the first protein coding gene, E1A, commencing approximately six hours after the infection of the primary lung fibroblasts [4]. E1A then orchestrates expression of all other viral transcriptional units and initiates the reprogramming of the infected cell to make it suitable for viral replication and growth [5,6]. E1A, the first protein expressed during viral infection, has the complex task of not only initiating the expression of all other viral genes, but is also charged with forcing the infected cell into the S-phase in order to enable viral genome replication. The primary way by which E1A carries out these duties is via binding to and altering the function of cellular proteins [5,6], as well as cooperating with viral proteins to drive efficient viral replication [7,8].

The *E1A* gene of HAdV5 encodes for five different splice variants that range in length from 289 amino acids found in the largest 13S mRNA-encoded splice variant, to 55 residues found in the smallest 9S mRNA-encoded protein [1]. The dynamics of these mRNAs, and consequently the proteins that are translated from them, vary substantially during the course of viral infection. Specifically, the two largest HAdV5 E1A isoforms (289R and 243R) are found predominantly early in the infection [9]. As the infection progresses, the smaller isoforms become more abundant, with the 10S-derived 171R protein becoming the most abundant detectable variant, while the two largest variants express at decreasing levels [9]. Interestingly, the differences between these isoforms and their relative contribution to viral infection have not been thoroughly studied. Surprisingly, neither has the contribution of the E1A regions outside of the conserved regions been thoroughly investigated in the context of viral fitness. We have recently used the *dl*1100 series of *E1A* mutants within the second exon of the protein to establish how these mutations affect overall viral replicative cycle [10], with surprising findings. For example, all E1A C-terminus mutants were deficient in growth, induction of S-phase, viral protein and gene expression, and their ability to replicate their genomes. Importantly, the mutants affected were not only the ones that deleted portions of the conserved region (CR) 4, but also those that deleted the inter-CR residues of the protein thought to play a less critical role in the viral life cycle [10]. These observations hinted at a potential important function of the regions outside of the CRs in viral replication and life cycle. We therefore hypothesized that we would see similar effects in deletion mutants within the first exon of *E1A* outside of the CRs.

In the present study we investigated the contribution that the region encoded by the first exon of *E1A* (excluding CR3) makes to the viral life cycle. We again used the *dl*1100 series of mutants spanning deletions of E1A from residues 4 up to 138 (Figure 1A). Surprisingly, many of these mutants have never been thoroughly investigated in their overall contribution to the viral fitness and our study provides the first comprehensive evaluation of these deletions on viral growth, gene and protein expression, genome replication, induction of S-phase, and deregulation of cellular cell-cycle specific genes. This study provides novel insights into E1A deletions outside of the CRs and provides greater understanding for those wanting to use adenoviral vectors for gene delivery, research, or treatment purposes.

## 2. Materials and Methods

### 2.1. Antibodies

Mouse monoclonal anti-E1A antibodies, previously described [11,12], and as used by us earlier [9,10], were grown in-house and used as the supernatant from the hybridoma cells. Mouse monoclonal anti-72kDa DBP antibody was previously described [13] and was used at a dilution of 1:400 for Western blot. Anti-adenovirus type 5 antibody was purchased from Abcam (catalog number ab6982, Cambridge, MA, USA). Actin antibody was purchased from Abcam (catalog number ab3280). Secondary antibodies were acquired from Jackson ImmunoResearch (West Grove, PA, USA) and were used at a dilution of 1:200,000.

### 2.2. Cell and Virus Culture

As carried out previously [9,10], 293 (ATCC CRL-1573) and WI-38 (ATCC CCL-7) cells were grown in Dulbecco’s modified Eagle’s medium (HyClone, Logan, UT, USA) and supplemented with 10% fetal bovine serum (VWR Seradigm, Mississauga, ON, Canada) and streptomycin and penicillin (HyClone). Cells were incubated at 37 °C with 5% CO_2_. All virus infections were carried out in serum-free media for 1 h, after which saved complete media was added without removal of the infection media. All viruses used were of the same genetic background as *dl*309, i.e., all mutants and *dl*309 had the same deletion in the E3 region. For all infections, titered crude freeze-thaw lysates were used.

### 2.3. EdU Incorporation Assay

As described previously [9,10], WI-38 cells were grown until 100% confluent on LabTek II 4-chamber slides (Thermo-Fisher, Waltham, MA, USA). After becoming fully confluent, cells were incubated for a further 72 h to achieve growth arrest. Infections were carried out as described above with a multiplicity of infection (MOI) of 100 for *dl*309 [14], or *dl*1101 through *dl*1109 [15]. One hour prior to fixation, cells were pulsed with EdU for 1 h as per manufacturer’s specifications using the Click-It EdU labeling kit for microscopy (Life Technologies, Carlsbad, CA, USA). After EdU labeling, cells were fixed in 3.7% formaldehyde, stained for EdU using the Click-It kit with AlexaFluor 488 and labelled for E1A using M58 monoclonal antibody and AlexaFluor 594 conjugated secondary anti-mouse antibody (Jackson ImmunoResearch). Cells were imaged using Molecular Devices (San Jose, CA, USA) ImageXpress Micro 4 high content imager and automatically analyzed using the MetaXpress software.

### 2.4. Immunofluorescence

As performed and described previously [10,16], WI-38 cells were plated at low density (40,000 cells per chamber) on chamber slides (Nalgene Nunc, Rochester, NY, USA) and subsequently infected as described above. Twenty-four hours after infection, cells were fixed in 4% formaldehyde, blocked in blocking buffer (1% normal goat serum, 1% BSA, 0.2% Tween-20 in PBS), and stained with specific primary antibodies. M58 was used neat (hybridoma supernatant) and AlexaFluor 488 secondary antibody (Jackson Immunoresearch) was used at a dilution of 1:600. After staining and extensive washing, slides were mounted using Prolong Gold with DAPI (Invitrogen, Carlsbad, CA, USA) and imaged using Zeiss LSM700 confocal laser scanning microscope. Images were analyzed using Zeiss ZEN software package.

### 2.5. PCR Primers

All primers used were previously described [9,16]. All primers were purchased from Integrated DNA Technologies and annealing temperature of 60 °C was used.

### 2.6. Real-Time Gene Expression Analysis

As performed previously [9,10], arrested WI-38 cells were infected with the different viruses at a MOI of 100 for 16, 24, 48 and 72 h. Total RNA was extracted using the NucleoZOL Reagent (Macherey-Nagel, Düren, Germany) at the indicated time points according to manufacturer’s instructions. An amount of 1.25 μg of total RNA was used in reverse-transcriptase reaction using SuperScript VILO reverse transcriptase (Invitrogen) according to the manufacturer’s guidelines using random hexanucleotides for priming. The cDNA was subsequently used for real-time expression analysis using the BioRad CFX96 real-time thermocycler (BioRad, Hercules, CA, USA). Analysis of expression data was carried out using the Pfaffl method [17] and was normalized to GAPDH mRNA levels; these were compared to *dl*309.

### 2.7. Statistical Analysis

Statistics were performed as described by [18] and as we have previously done [10]. Briefly, statistical analysis was conducted using one-way analysis of variance (ANOVA) followed by post hoc comparison using Tukey test of cellular and viral genes from *dl*309 infection versus mutant virus infection. Viral growth assays and genome quantification assays were also subjected to ANOVA with post hoc comparison using Tukey test comparing mutants to *dl*309. *p*-values were two-tailed and values of <0.05 were considered statistically significant in gene expression assays, viral growth assays and genome quantification assays. Student’s independent sample *t*-test was conducted on EdU incorporation assays. *p*-values were one-tailed and values of <0.05 were considered statistically significant in the EdU incorporation assays.

### 2.8. Viral Genome Quantification

As we have done previously [9,10], arrested WI-38 cells were infected with the different viruses at a MOI of 100. The cells were infected for 48 or 72 h and were lysed in lysis buffer (50 mM Tris pH 8.1, 10 mM EDTA and 1% SDS) on ice for 10 min. Lysates were sonicated briefly in a Branson 450 Analog Sonifier to break-up cellular chromatin and subjected to digestion, using Proteinase K (NEB) according to manufacturer’s specifications. Following digestion, viral DNA was purified using GeneJET PCR Purification Kit (Thermo-Fisher). PCR reactions were carried out using SYBR Select Master Mix for CFX (Applied Biosystems, Foster City, CA, USA) according to manufacturer’s directions using 2% of total purified DNA as template using a CFX96 Real Time PCR instrument (BioRad). Standard curve for absolute quantification was generated by serially diluting pXC1 plasmid containing the left end of HAdV5 genome starting with a concentration of 1.0 × 10^7^ copies per reaction down to 1.0 copy per reaction. The cellular interferon β1 gene was used to quantify cellular genomes. The primers used were the same as those used for expression analysis of E1B region; the annealing temperature used was 60 °C; and 40 cycles were run.

### 2.9. Virus Growth Assay

As previously described [9,10], arrested WI-38 cells were infected with the different viruses at MOI of 100 in serum-free medium. The mutant *dl*309 [14] expressing wild-type (wt) E1A is deleted for E3 14.7K, 14.5K, and 10.4K proteins [19] and does not express VAI RNA [20], and all of the mutants used were generated in *dl*309 background making comparisons directly to *dl*309 most appropriate. Importantly, in cell culture models, *dl*309 grows as well as wt virus [14,20,21]. Virus was adsorbed for 1 h at 37 °C under 5% CO_2_, after which cells were bathed in conditioned media and were re-incubated at 37 °C under 5% CO_2_. Virus titers were determined at 48, 72 and 96 h after infection by plaque assays performed on 293 cells by serial dilution.

## 3. Results

### 3.1. Deletions in the First Exon of E1A Affect Virus Growth

We have previously [10] investigated the effects of deletions within the second exon of HAdV5 *E1A* on overall viral fitness. We therefore wanted to extend this study to the region of E1A encoded by the first exon outside of the well-studied CR3. We have utilized a series of N-terminus E1A deletions, the so-called Bayley mutants generated in the lab of Stan Bayley in the late 1980s [15,22,23,24,25,26,27], to investigate how short deletions of E1A affect viral fitness. Firstly, we set out to determine how growth of these mutants is affected in comparison to the wt E1A-expressing mutant *dl*309 [14] in primary and contact-inhibited human lung fibroblasts WI-38 [28]. We chose these cells and their arrested state as they most closely represent the natural target of the human host and their growth arrested status will pose the most significant challenge to the virus, allowing any defects in growth to be identified more readily. All mutants showed growth and growth kinetics that differed from *dl*309 expressing wt E1A proteins (Figure 1B). Unexpectedly, some mutants showed growth that was higher than what was observed with *dl*309. For example, the mutant *dl*1106 grew to significantly higher titers than *dl*309 at all time points investigated. However, most mutants ultimately showed growth defects as compared to *dl*309 even when their initial titers were higher, such as *dl*1102—that had higher titers at 48 and 72 h after infection, but lagged slightly behind *dl*309 at 96 h after initial infection. Other mutants showed severe growth defects, in particular *dl*1103 lagged by nearly 2 logs at all time points behind *dl*309, while *dl*1109 showed a 1 log growth deficit at all time points in comparison to *dl*309. Overall, these results show that there exist substantial differences in the ability of E1A first exon 1 deletion mutants in support of virus growth in contact inhibited primary lung fibroblasts.

We have also investigated the appearance of the cytopathic effect (CPE) during viral growth (Figure 2). Generally, the appearance of CPE is correlated with virus growth, with *dl*309 showing observable changes to cellular morphology at 48 h after infection. For example, mutants *dl*1102 and *dl*1106 that grew similarly to or better (as was the case with *dl*1106) than *dl*309 showed extensive CPE by the end point of the assay (96 h after infection) while mutants that grew poorly did not, such as *dl*1103 and *dl*1104.

### 3.2. Deletions within Exon 1 of E1A Do Not Affect Its Sub-Cellular Localization

E1A has previously been reported to contain a second nuclear localization signal (NLS) within the first exon of the gene [29]. We therefore wanted to determine whether deletions within this exon affect the sub-cellular localization of E1A while its strong C-terminal NLS remains intact during viral infection. All E1A deletion mutants showed strong nuclear localization (Figure 3) with similar morphologies. We did not observe detectable levels of E1A in the cytoplasm.

### 3.3. Viral Genome Replication in Viruses Expressing E1A Deletion Mutants

Deletions within the first exon of *E1A* affect a wide variety of both viral and cellular processes that can influence viral replication. One key process is the induction of S-phase in the infected cell that directly impacts the ability of the virus to replicate its genome, other effects will directly affect expression of viral proteins necessary for genome replication, such as the DNA polymerase. We therefore investigated how the deletions within the first exon of *E1A* affect the ability of the virus to replicate its genome (Figure 4). Replication of viral genomes was assessed at 48 and 72 h after infection. This represents a time after the onset of genome replication, which we have previously established to occur at approximately 30 h after infection, in primary arrested lung fibroblasts [4]. Several of the mutants studied showed severe deficiencies in their ability to replicate their genomes, including *dl*1101, *dl*1102, and *dl*1103. These mutants showed deficiencies at all times analyzed. Other mutants showed more moderate defects, especially at 48 h after infection, including *dl*1108 and *dl*1109. Interestingly, some mutants replicated their genomes more efficiently than the virus expressing wt E1A and these were mutants *dl*1105 and *dl*1106. Overall, our results show that deletions at the extreme 5’ end of the first exon of *E1A* (N-terminus and CR1 of the protein) have the most dramatic effect on viral genome replication followed by mutants within CR2, while deletions in the inter-CR region appear to enhance genome replication particularly at late times in infection.

### 3.4. E1A Exon 1 Mutants Affect Viral Gene and Protein Expression

E1A is central to the regulation of expression of all viral genes after infection and it is found on all viral promoters during the course of infection [30,31]. We therefore investigated how the effects of the deletions within the first exon of *E1A* affect its ability to transactivate and regulate viral promoters. We examined viral gene expression at 16, 24, 48, and 72 h after infection and compared it to the expression of these genes from cells infected with *dl*309, expressing wt E1A (Figure 5). Early in infection, at 16 h, all viruses showed viral gene expression that was lower than that observed in *dl*309-infected cells with the exception of mutants *dl*1102 that showed most genes being slightly higher and *dl*1109 that showed similar gene expression to *dl*309. This trend continued at 24 h after infection, although at this time-point we observed smaller differences with the mutants that were deficient as compared to *dl*309, with some mutants now having higher gene expression of some but not all genes. In particular, mutant *dl*1106 showed elevated gene expression of the late gene as well as the viral DNA polymerase. At 48 h after infection we observed that most mutants, with the exception of *dl*1101, *dl*1103, and *dl*1104, showed viral genes to be modestly upregulated as compared to *dl*309 and this trend was also observed at 72 h after infection.

We also investigated how E1A deletions affect viral protein expression (Figure 6). We observed highly variable E1A levels between the different mutants at all times during infection, with some mutants having very low or undetectable levels of the protein, particularly at the earlier time points. Interestingly, the levels of E1A mRNA did not necessarily correlate with that of the observed protein. For example, levels of E1A mRNA were approximately 5-fold higher in *dl*1108 and *dl*1109 mutants than those in *dl*309 at 72 h after infection yet the E1A protein levels were much lower. Intriguingly, we also observed substantial variability in the stoichiometry of different E1A isoforms across the mutants. This was particularly apparent with *dl*1105 that expressed nearly equal levels of the four detectable E1A proteins at 24 h after infection, while later expressing only the 12S and 10S-derived polypeptides. Nevertheless, the 10S-derived 171R and equivalent mutant variants was generally the most abundant E1A isoform detectable later in infection, as we have observed before [9,10]. Generally, levels of the 72kDa DNA binding protein (DBP) correlated with the levels of E1A, but did not correlate with viral genome replication. Whereas the levels of late and structural proteins did not correlate with E1A or DBP levels, but rather followed the pattern of viral genome replication. Interestingly, the concentrations of structural proteins did not directly correlate with growth, as, for example, the mutant *dl*1102 had lower protein levels than *dl*309 yet was capable of similar growth. We did not detect any late proteins at 24 h after infection in all infected samples (not shown). In summary, our results show dramatic differences between viral gene and protein expression across the different mutants of E1A.

### 3.5. Induction of S-Phase and Expression of Cellular S-Phase Specific Genes by Exon 1 E1A Mutants

HAdV relies heavily on the host cell to replicate, requiring the infected cells to efficiently enter the S-phase in order to enable viral genome replication [1]. We therefore investigated the ability of the virus to drive the infected cells into S-phase (Figure 7) and its ability to deregulate expression of a small set of S-phase specific genes, required for cellular genome replication (Figure 8). All mutant viruses, with the exception of *dl*1102, were deficient for induction of S-phase as compared to *dl*309. Interestingly, all viruses were capable of driving cells into S-phase as compared to arrested cells that were mock-infected. Surprisingly, viral mutants that are unable to bind to the Retinoblastoma (pRb) family of proteins were still relatively adept at driving S-phase (mutants *dl*1107 and *dl*1108) in arrested cells. The ability of the different mutant viruses to replicate was not correlated with their ability to drive the S-phase, neither was the overall viral genome replication. For example, the mutant *dl*1106 was quite poor at driving S-phase at 24 h after infection but was one of the better viruses in terms of its growth and capacity to replicate its genomes. In fact, it was similar in those two aspects to *dl*1101, which grew poorly and was severely deficient for its ability to replicate its genomes. In the end, most mutants showed differences and deficits in their ability to drive S-phase but this had little correlation to the ability of the virus to replicate.

The ability of the virus to replicate its genome is more likely due to the cell cycle state of the cell rather than the virus itself. We therefore investigated the ability of the mutants to deregulate a set of cellular S-phase specific genes (Figure 8). Overall, most viruses with the exception of *dl*1102 were deficient, as compared to *dl*309, at early time points in their ability to induce expression of *MCM4, BLM,* and *PCNA.* This deficit largely continued at 24 h but some viruses were approaching expression levels observed in *dl*309-infected cells, including *dl*1101, *dl*1105, *dl*1106, and *dl*1109. After 48 h, we observed little difference between the expression of cellular genes in the mutant-infected cells and those infected with *dl*309 expressing wt E1A. In conclusion, our results show that the most pronounced defects were observed early in infection and—as the infection progressed—both viral and cellular genes were able to equalize or surpass expression levels observed in *dl*309-infected cells.

## 4. Discussion

The present study investigated the effects that small deletions within the first exon of HAdV5 *E1A* gene (Figure 1A) have on the virus replicative cycle in arrested human lung fibroblasts, WI-38. Prior studies of these mutants have focused on a narrow aspect of their properties, mainly focusing on either transactivation or transformation. Our study is the first comprehensive analysis of overall viral fitness that is affected by these deletions. Importantly, our results show clear differences in how these mutations affect virus growth, DNA replication, protein expression, and modulation of the cell cycle. Significantly, we identified several mutants that exhibited characteristics that may be of interest to researchers using adenoviruses as vectors for gene delivery or other purposes, such as conditionally replicating viruses or oncolytic agents. Together with our earlier study [10], we provide a comprehensive view of the overall fitness of the Bayley E1A mutants in the course of the viral replicative cycle. Table 1 summarizes the results of our study, together with earlier studies of these mutants.

Our growth results show some intriguing characteristics of the effects the different E1A mutants have on viral replication (Figure 1B). Although the virus expressing wt E1A (*dl*309) showed the best growth at the end of our assay, with the exception of *dl*1106, it was slower to attain highest replication than certain mutants, such as *dl*1102, *dl*1107, and *dl*1108. Deletion in *dl*1106 removes amino acids 90-105 [15], which are located between CR1 and CR2. This deletion has not been reported to have significant effects on the viral properties previously investigated, including binding to a subset of cellular proteins and induction of transformation in cooperation with activated *ras* [22]. For example, this mutant is still able to bind the pRb family of proteins and p300/CBP [22,35], as expected based on the position of the deletion. However, this mutant was deficient in its ability to induce S-phase (Figure 7) as compared to *dl*309. In fact, it was the worst in its ability to drive S-phase, only able to induce it in approximately 20% of the infected cells, compared to 60% observed for *dl*309 24 h after infection. This deficiency in S-phase induction was even worse than mutants *dl*1107 and *dl*1108, which have compromised binding to the pRb family of proteins [22]. The deletion found in *dl*1106 will reduce the spacing between CR1 and CR2, which may affect the ability of this mutant to disrupt interactions between pRb and the E2F family of proteins, explaining potential deficiency in driving S-phase. Yet, this does not explain the overall efficiency in virus growth, genome replication, and protein expression as compared to other mutants. Importantly, these results demonstrate that regions between CRs of HAdV5 E1A have important but not yet fully characterized functions, something we also observed when investigating exon 2 of *E1A* in the viral replicative cycle [10]. 

Induction of S-phase will be dependent on the activation of various S-phase specific genes either through the disruption of the E2F family of factors or via other mechanisms. Most of the viruses that we investigated were deficient in activation of S-phase specific genes early on during the infection, but later these viruses were able to reach expression levels similar to what was observed with the wt E1A expressing *dl*309. This early deficiency correlated with the EdU incorporation results indicative of S-phase induction. It is likely that the presence of the largest E1A isoform, 289R, that is by itself directly able to drive S-phase specific genes [36], contributes to the less than expected effect of pRb-deficient mutants in their ability to grow and replicate. In order to fully block the ability of E1A to drive S-phase, a triple mutant would be necessary that deletes the E2F/DP-1 binding domain located at the N-terminus, as well as the CR1 and CR2 pRb binding domains [37], which we did not investigate. Nevertheless, some of the mutants that showed defects in driving S-phase and induction of DNA replication genes were able to make up the deficit in gene expression later in the replicative cycle and ended up replicating comparably to wt E1A-expressing virus. This is reminiscent to what was previously observed with E1A complementation for a production cell line based on A549 lung adenocarcinoma cells [38]. In that study, it was observed that complementation of E1-deleted viral vectors led to highest virus growth using a double deletion mutant of E1A, *dl*1101/*dl*1107 [38], while cell lines expressing wt E1A were poor at complementing viral growth due to early apoptosis. Although, despite the lack of E1B-19k complementation, it was suggested that wt E1A expression in these cells causes early apoptosis following infection. Apoptosis was not observed in the double-mutant expressing cell line, likely due to reduced induction of pro-apoptotic E2F-regulated genes [39,40], which may be overactive in wt E1A-expressing cells. This agrees with later studies, showing that E1A289R can directly drive E2F-regulated gene expression via binding to E2F/DP-1 complexes on cellular promoters, in addition to its function of disrupting pRb-E2F complexes [36].

Our results show a direct correlation between viral genome replication, as quantified by viral genome copies per cellular genome (Figure 4), and the expression of viral late proteins, as visualized by a Western blot for structural proteins (Figure 6). Although this is not a new finding [1,41], what is intriguing about this observation is that late gene expression of transcripts beyond the L1 polyadenylation signal has recently been reported to start as early as 13 h after infection in arrested lung fibroblasts [4]. Since there are no late proteins detectable at 24 h after infection (Figure 6), this suggests that either translation of these transcripts occurs at a very low level or there is some sort of translational inhibition of late transcripts. One might speculate that this could occur either via restricted export of these mRNAs into the cytoplasm or some type of other inhibitory mechanism. The former is a likely possibility as E1B-55k and E4orf3 are known to play a role in late mRNA export during infection [42,43,44]. Interestingly, we also observed altered stoichiometry of viral structural proteins (Figure 6), which may contribute to the ability of various mutants to assemble functional capsids. For example, the ratio of protein V and VI to hexon and penton is much different in some mutants (particularly *dl*1105 and *dl*1106), this may alter the efficiency of capsid assembly leading to lower viral titer. Curiously, we observed very little correlation between genome replication and the levels of the E2-72kDa DNA binding protein (DBP; Figure 6). For example, the mutant *dl*1101 expressed high levels of DBP, comparable to wt, replicated genomes very poorly, while the mutant *dl*1105 expressed low levels of DBP yet replicated its genomes to one of the highest levels observed. Other mutants showed a correlation between low genome copies and low DBP levels, such as *dl*1103. This suggests that a certain amount of DBP is required for efficient genome replication, beyond which no positive effect is observed. This is logical to assume since DBP is only an accessory factor required for genome replication, which will be carried out by the viral DNA polymerase in cooperation with the pre-Terminal Protein and cellular factors, which likely play a much bigger role.

The E1A mutant *dl*1103, which deletes residues 30–49 that includes a portion of the CR1, was the mutant that grew the poorest and showed overall reduced fitness in all other aspects investigated. This deletion affects the ability of this mutant E1A to efficiently bind to p300/CBP and the pRb family of proteins [22], p400 [45], drive mitosis [45], as well as cooperate with activated *ras* in transformation of primary rat kidney cells [22]. In addition, we observed that this mutant expressed poorly and was ineffective in induction of DBP protein expression and viral late proteins (Figure 6), as well as induction of viral genes. Interestingly, its ability to drive genome replication was closest to that of *dl*1102, which replicated over 100-fold better. Some of the other aspects of the overall fitness of *dl*1103 were similar to other mutants that grew much better, such as *dl*1101 and *dl*1102. This suggests that there are still many aspects of E1A that we do not fully understand, in terms of its ability to support viral replication and reprogram the infected cells. The other poorly growing mutant, *dl*1109, is more enigmatic in terms of the reasons as to its overall poor performance. This mutant binds to all the proteins that *dl*1103 fails to bind [22,45], but it is defective for transformation with activated *ras* [22,33]. Unlike *dl*1103, *dl*1109 was much better at replicating its genomes and driving late protein expression, it was also similar to wt E1A in terms of its ability to drive viral gene expression. Therefore, its overall poor growth is difficult to reconcile with overall viral fitness and likely has more to do with the cell state and the ability of this mutant to reprogram the infected cell, rather than viral-intrinsic factors. Nevertheless, these results further highlight the complex nature of the relationship between E1A and the host, and much remains to be investigated to fully understand this relationship.

The current study investigated the effects of *E1A* deletion mutants in the first exon of the gene outside of the CR3 in terms of overall viral fitness in growth-inhibited primary lung fibroblasts WI-38 cell culture model system. Although our results examine a wide array of viral characteristics, it should be noted that the main limitation of our investigation is its use of a cell culture model system, rather than a live organism. This limitation is largely due to lack of a suitable model animal for human adenovirus research, and as such cell culture models are well accepted in the field. Our results highlight the great diversity in the overall fitness of these different E1A mutants in viral growth, gene expression, genome replication, and host reprogramming. Importantly, we identify several novel aspects of these mutants and their relationship to overall fitness. These studies provide a comprehensive overview of the influence of E1A on viral replicative cycle and may provide useful insight to those using adenovirus for gene delivery, treatment, or oncolytics. Our results also highlight the complex nature of the interaction between the host and E1A, paving the way for future studies using these invaluable mutants.

## Figures and Tables

**Figure 1 viruses-12-00213-f001:**
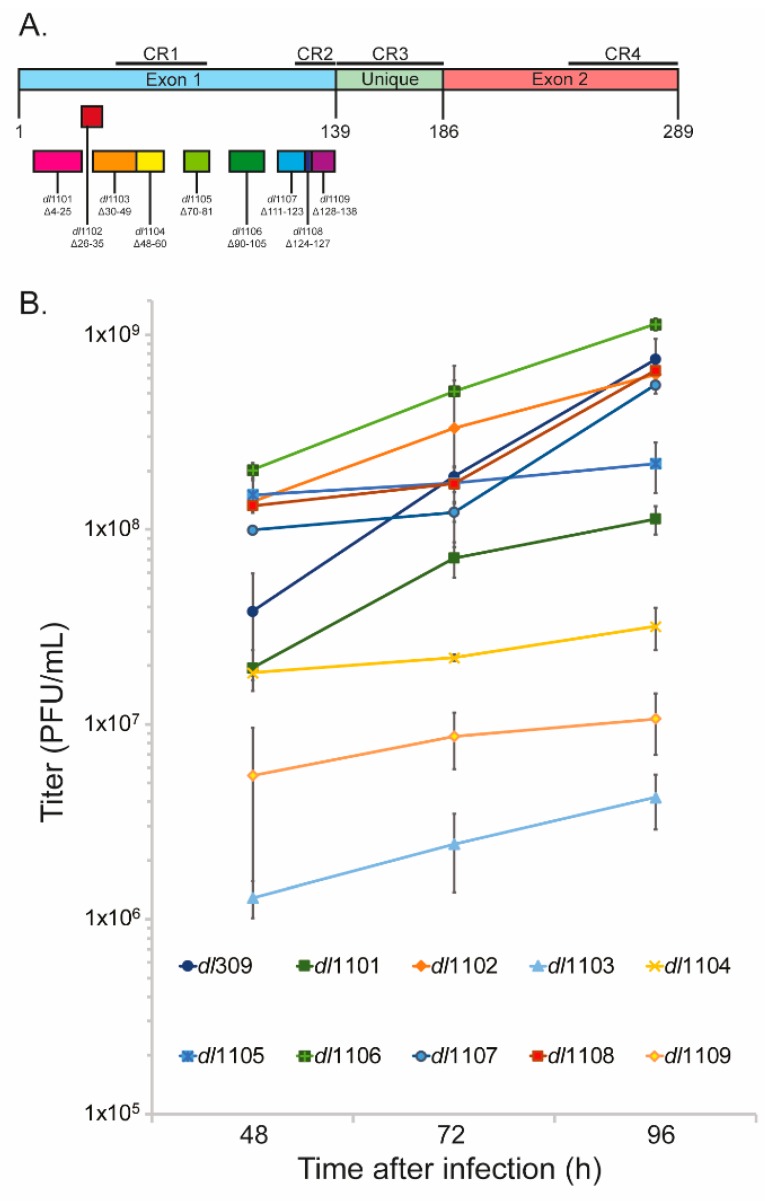
Deletions within E1A exon 1 affect virus growth in primary lung fibroblasts. (**A**) Schematic representation of E1A mutants used in this study. (**B**) WI-38 cells were grown until fully confluent at which point medium was replaced and cells were incubated for 72 h before infection with the indicated viruses at a multiplicity of infection (MOI) of 100. Virus growth was subsequently evaluated by harvesting cells at the indicated time-points, followed by 3 rounds of freeze-thaw to liberate viral particles, and titration on 293 cells by plaque assay. Error bars represent standard deviation of three biological replicates. *p*-values for differences between given time points is as follows: 48 h, all were significant versus *dl*309 with a *p-*value of <0.0001; 72 h, *dl*1102 and *dl*1108 were not significant with regards to *dl*309, all others were with a *p-*value of ≤0.0329; 96 h, all were significant with a *p-*value < 0.0001.

**Figure 2 viruses-12-00213-f002:**
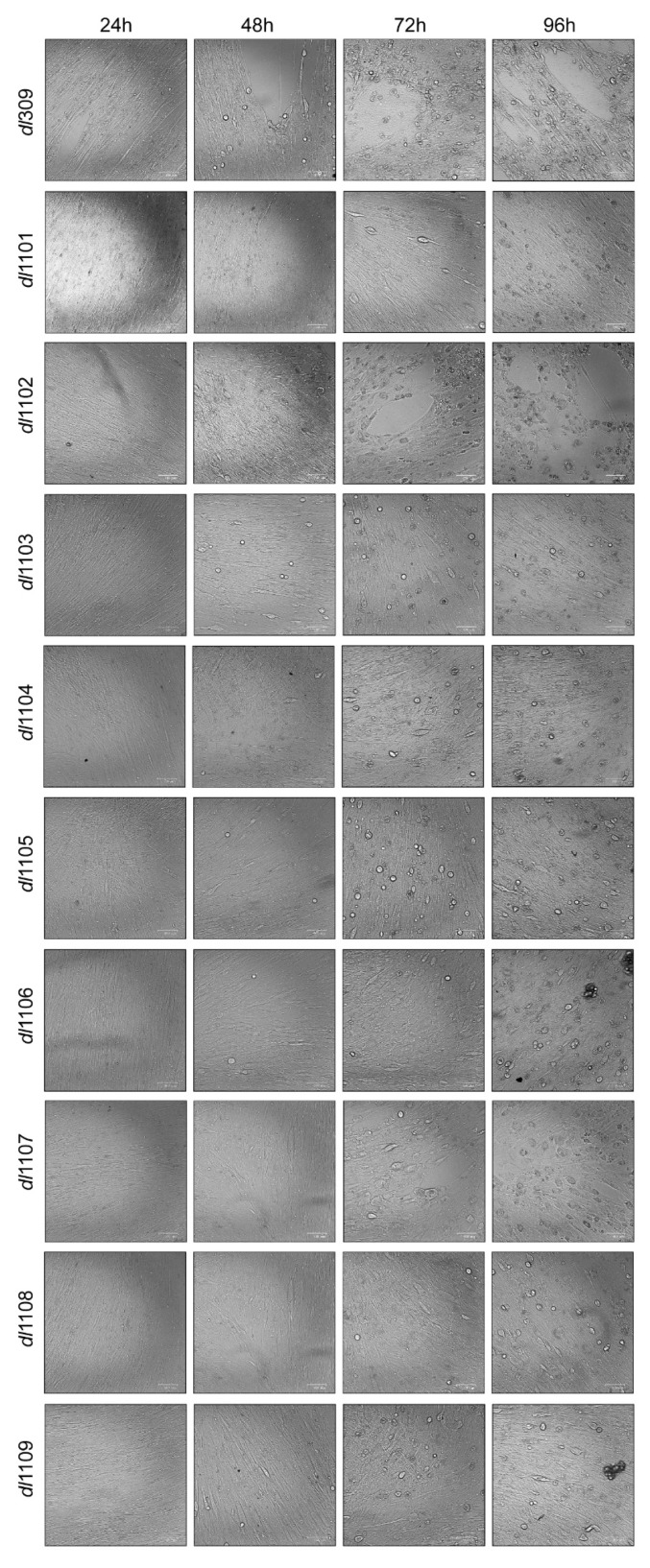
Cytopathic effect in exon 1 E1A harboring HAdV-infected WI-38 cells. WI-38 cells were grown until confluence at which point medium was replaced and cells were incubated for 72 h before infection with the indicated viruses at a MOI of 100. At the indicated time points cells were imaged in the bright field using the BioRad ZOE cell imager.

**Figure 3 viruses-12-00213-f003:**
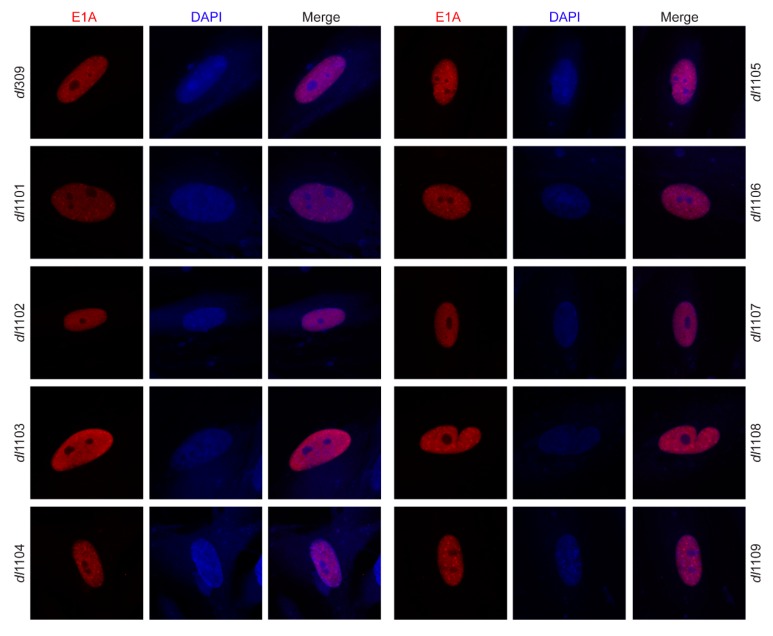
E1A mutant sub-cellular localization in primary lung fibroblasts WI-38. WI-38 cells were grown until confluence at which point medium was replaced and cells were incubated for 72 h before infection with the indicated viruses at a MOI of 100. Twenty-four hours after infection cells were fixed and stained for E1A (red) using the M73 anti-E1A antibody. DAPI was used as a nuclear counterstain. Images were then acquired on a Zeiss LSM700 laser confocal microscope using the 63× objective lens.

**Figure 4 viruses-12-00213-f004:**
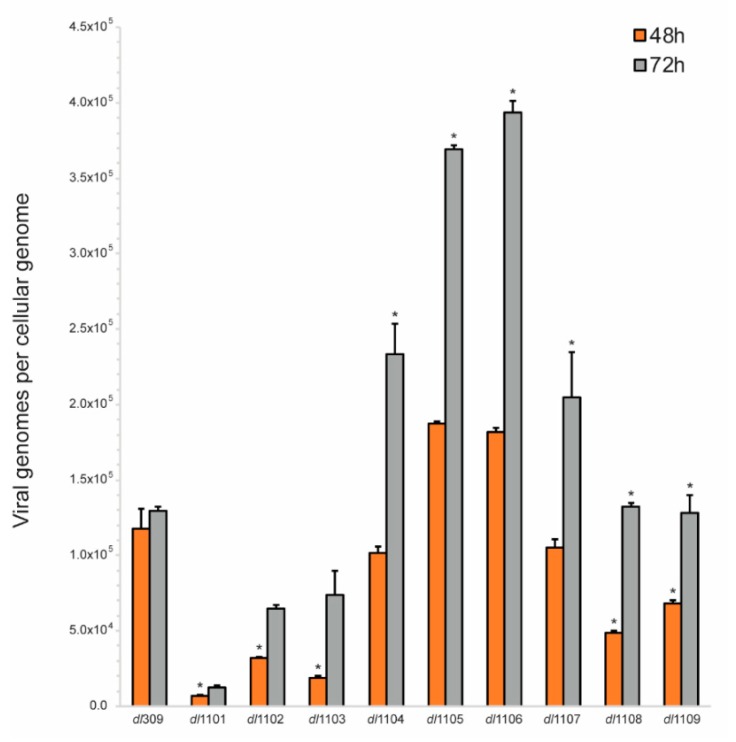
Viral genome replication in exon 1 E1A mutant viruses. WI-38 cells were grown until confluence at which point medium was replaced and cells were incubated for 72 h before infection with the indicated viruses at a MOI of 100. At the indicated time-points viral genomes were quantified using quantitative real-time PCR using SYBR Select Master Mix for CFX and normalized to cellular genomes using the interferon β1 gene. Data are represented as viral genomes per cellular genome from three biological replicates. Error bars represent standard deviation of biological replicates. Asterisk denotes differences from *dl*309 that are statistically significant with a *p* value of <0.0001.

**Figure 5 viruses-12-00213-f005:**
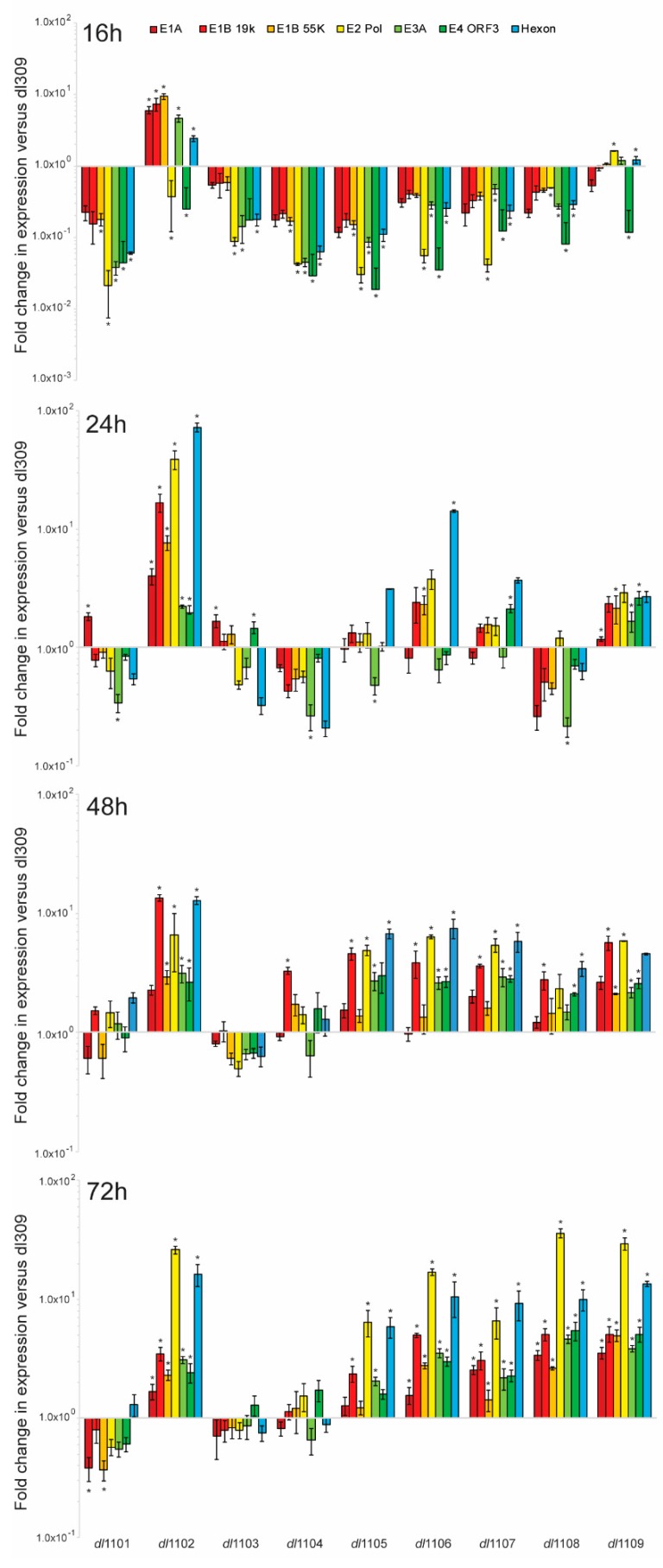
Expression of viral genes from E1A exon 1 mutants relative to *dl*309 in arrested WI-38 cells. WI-38 cells were grown until confluence at which point medium was replaced and cells were incubated for 72 h before infection with the indicated viruses at a MOI of 100. At the specified time-points total cellular RNA was extracted using NucleoZOL according to the manufacturer’s protocol followed by DNase I treatment to remove contaminating cellular and viral DNA. cDNA was synthesized using SuperScript VILO Master Mix and quantitative real-time PCR was performed on a BioRad CFX96 real-time thermocycler using SYBR Select Master Mix for CFX. Data were analyzed using the Pfaffl method and is presented as fold change versus *dl*309. Error bars represent standard deviation of three biological replicates, asterisks represent statistically significant changes with a *p-*value of <0.0001.

**Figure 6 viruses-12-00213-f006:**
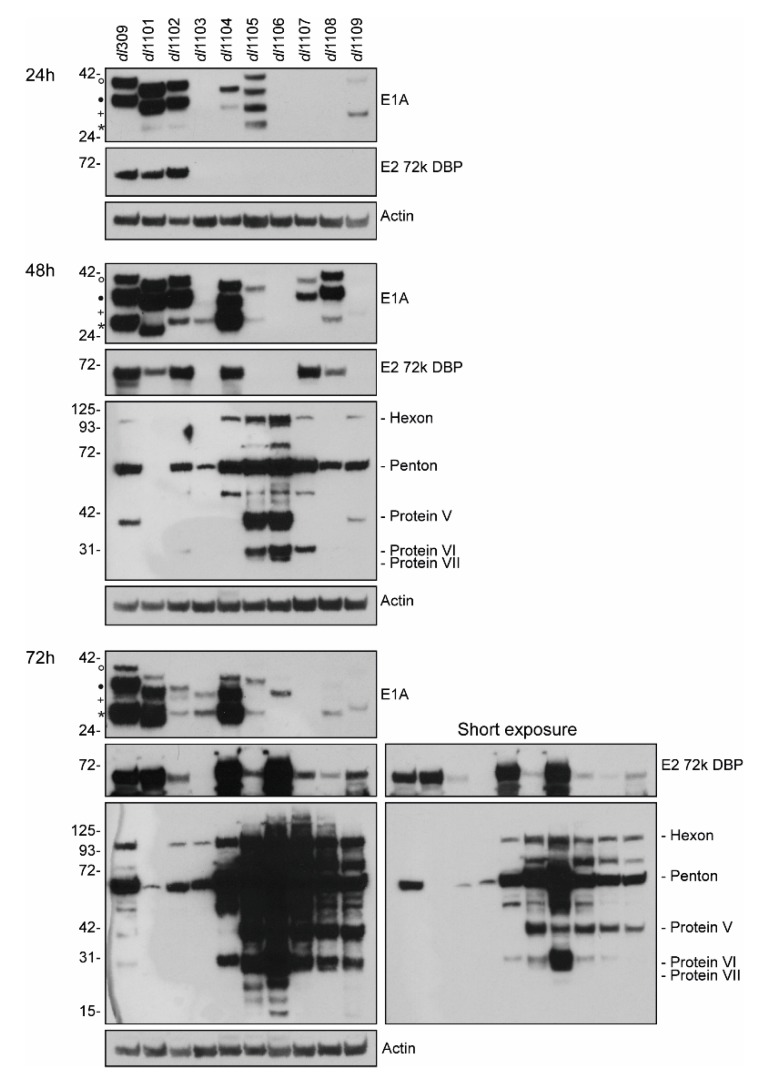
Viral protein expression in WI-38 cells infected with HAdV5 E1A mutants within exon 1. WI-38 cells were grown until confluence at which point medium was replaced and cells were incubated for 72 h before infection with the indicated viruses at a MOI of 100. At the specified time-points cells were harvested and proteins extracted using NP-40 lysis buffer as described in materials and methods. Twenty-five micrograms of total cell lysate was resolved per virus on SDS-PAGE using the Novex BOLT 4–12% gradient gels using the MES buffer. Proteins were subsequently transferred to a PVDF membrane using Genscript’s eBlot L1 with default settings, and the membranes were blotted for the indicated proteins. Detection was performed using secondary HRP-conjugated antibodies and exposure to radiographic film. ◦ denotes 13S-derived 289R, • denotes 12S-dervied 243R, ^+^ denotes 11S-derived 217R, and * denotes 10S-derived 171R.

**Figure 7 viruses-12-00213-f007:**
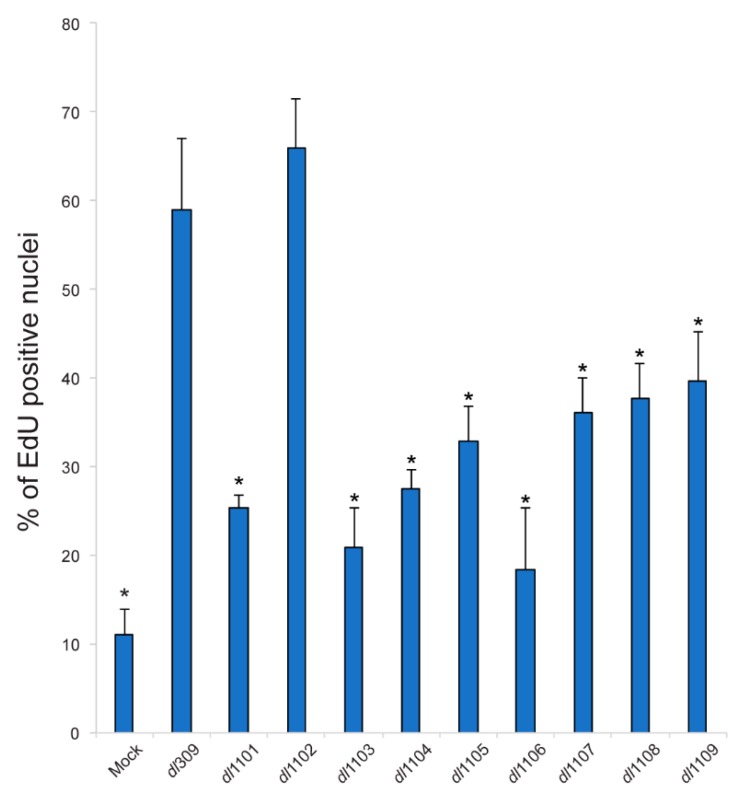
Induction of S-phase by E1A exon 1 mutants. WI-38 cells were grown in a four-chamber LabTek II chamber slides until confluence at which point medium was replaced and cells were incubated for 72 h before infection with the indicated viruses at a MOI of 100. At 23 h after infection, cells were pulsed for 1 h with EdU and then fixed with paraformaldehyde and stained using the Click-It EdU labeling kit for microscopy. After EdU labeling, cells were stained for E1A to mark infected cells using the M73 monoclonal antibody and Alexa 594-conjugated secondary anti-mouse antibody. DAPI was used as a nuclear counterstain. After staining, slides were mounted and imaged using Molecular Devices ImageXpress Micro 4 and analyzed using MetaXpress software. At least five independent fields of view were acquired per virus, with a minimum of 100 cells counted per field of view. Error bars represent standard deviation of five independent fields of view, asterisks represent statistically significant changes *versus dl*309-infected cells with a *p-*value of <0.0001.

**Figure 8 viruses-12-00213-f008:**
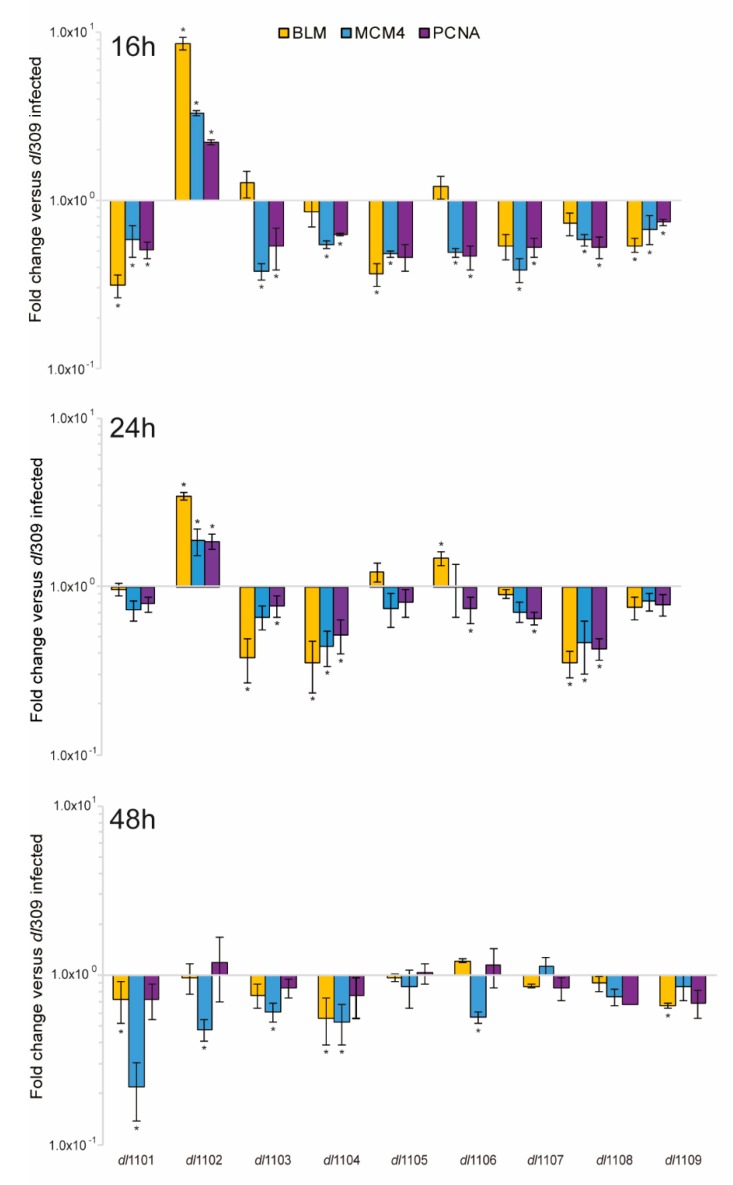
Expression of cellular S-phase regulated genes in WI-38 cells infected with E1A exon 1 HAdV5. WI-38 cells were grown until confluence at which point medium was replaced and cells were incubated for 72 h before infection with the indicated viruses at a MOI of 100. At the specified time-points total cellular RNA was extracted using NucleoZOL according to the manufacturer’s protocol followed by DNase I treatment to remove contaminating cellular and viral DNA. cDNA was synthesized using SuperScript VILO Master Mix and quantitative real-time PCR was performed on a BioRad CFX96 real-time thermocycler using SYBR Select Master Mix for CFX. Data were analyzed using the Pfaffl method and is presented as fold change versus *dl*309. Error bars represent standard deviation of three biological replicates, asterisks represent statistically significant changes with a *p-*value of ≤0.0208.

**Table 1 viruses-12-00213-t001:** Summary of characteristics of E1A exon 1 and 2 deletion mutants. The number of plusses (+) indicates how strong the given effect is, a minus (−) indicates that this was not observed, a (+/−) indicates a very weak effect. N—nuclear, C—cytoplasmic.

Strain	Viral Growth	CPE	Subcellular Localization	Viral Genome Replication	Viral Gene Expression	Induction of S-phase	S-phase Gene Expression	Transformation with Ras ^b^	pRb ^b^ Binding	p300 ^b^ Binding
***dl309***	+++	++++	N	+++	++++	++++	++++	++++	+++	+++
***dl311*** ^a^	+	+/−	N, C	+	+	+	+/−	++	++	+++
***dl1101***	++	+	N	+	+	+	+	+/−	++	−
***dl1102***	+++	++++	N	++	++++	++++	++++	+++	++++	++++
***dl1103***	+	+	N	++	+	+	+	+	+	+
***dl1104***	++	+	N	+++	++	++	+	+	+	−
***dl1105***	++	++	N	++++	+++	++	++	+++	++	++
***dl1106***	++++	+++	N	++++	+++	+	++	++++	+++	++++
***dl1107***	+++	++	N	+++	+++	++	++	+	−	++++
***dl1108***	+++	++	N	++	++	++	++	−	−	+++
***dl1109***	+	+	N	++	++++	++	++	+	++	++++
***dl1116*** ^a^	+++	++	N	++	+	++	+++	++++	++	+++
***dl1132*** ^a^	+++	+	N	+	++	+	+	++	++	+++
***dl1133*** ^a^	+++	+++	N	+	+++	++	++	+++	++	+++
***dl1134*** ^a^	++	+++	N, C	++	+	++	+++	++++	++	+++
***dl1135*** ^a^	+++	+++	N, C	+	++	++	++	++++	++	+++
***dl1136*** ^a^	+	+	N, C	++	++	++	+/−	+++	++	+++

^a^ Based on reference [10]. ^b^ Based on references [14,22,32,33,34].

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
