# Peer review of "Characterization of Adenovirus 5 E1A Exon 1 Deletion Mutants in the Viral Replicative Cycle"

_viruses, 2020, doi:10.3390/v12020213_

Round 1

Reviewer 1 Report

In this study, the authors perform an extensive characterization of the ability of various E1A exon mutant viruses to: 1)induce S-phase in host cells 2)induce transcription of early and late viral genes, 3)induce host S-phase associated genes, 4)produce E1A and E2 as well as late structural proteins, 4)replicate their genomes, and 5) ultimately produce new infectious viral particles.  All of this was performed in the context of in vitro infections of WI-38 human lung fibroblasts.

This is the first study to my knowledge that performs all these experiments on this series of these dl1100 series of mutants.  The experimental design is sound and the conclusions are supported by the results presented.

I believe these results will be of interest to readers of Viruses and recommend the paper be accepted with some minor revisions which I suggest below:

1) Figure 2 is not very clear the way it is presented.  The images are blurred and too small.  Either remove the figure and refer to CPE as data not shown in this case, or make modifications so the figure is larger, maybe takes up a whole page or show less time-points and make the images bigger.  Some of the images are too dark as well and could use some contrast enhancement.

2) Figure 5 has text that is too small to read without a magnifying glass.  I suggest converting that figure into a full page - similar to what was done with Figure 8.

3) In figure 6, there was also a big difference in the relative levels of structural proteins compared to wildtype.  For example, the ratio between proteins IV and V to penton or hexon varies quite a bit in the deletion mutants and does not seem to resemble the wildtype protein.  Viral packaging and assembly is most likely dependent on the stoichiometric ratios of these proteins in addition to whether or not they are expressed.  This could be discussed.

4) One drawback to this study is that infections are performed in vitro and thus isolated from the selective weight that a fully functional immune system would have.  It is possible that in the context of a physiological infection, delayed expression of viral genes or inappropriate expression levels of these proteins could prove very detrimental to the fitness of the virus.  This limitation of the study should be mentioned in the discussion.

Reviewer 2 Report

A number of years ago, Stan Bayley constructed a series of mutants with small deletions in the “first exon” of the Ad5 E1A protein.  These “Bayley” mutants were characterized for their ability to immortalize cells and to transform cells in cooperation with activated Ras.  These mutants have been studied for their transforming abilities by many labs.

The Bayley mutants were built into a dl309 background, i.e. they are replication-competent on 293 cells, a cell line that provides E1A proteins in trans.

The goal of this current manuscript is to characterize the ability of these Bayley mutants to replicate in primary contact-inhibited human lung fibroblasts (WI-38 cells).  Numerous assays were conducted to assess: growth, CPE, subcellular localization of mutant proteins, burst size, expression of viral genes relative to “wild type” dl309, expression of E1A and other viral proteins by western blot, induction of cellular S-phase, and induction or repression of certain S-phase-regulated cellular genes.

The experiments seem to be well conducted and much data have been obtained.  One problem is what is the “take home message” from all the data.  The E1A proteins in particular are expressed very differently by the various mutants: some abundantly, and some at low levels.  Nevertheless, this is a study well worth doing, and will provide a basis for further analysis of these E1A mutants on aspects of viral replication.

One point to note:  all the studies were done in a dl309 background.  dl309 is not wild type Ad5 – it has a large deletion in the E3 region that precludes synthesis of the E3 proteins ADP (E3-11.6k), RIDα (10.4k), RIDβ (14.5k), and (I think) E3-14.7k.  ADP is known to have a profound effect on viral growth (ADP mutants have small plaques).  Lack of ADP could affect the results in this current study.  So could the lack of the other E3 proteins.  As such, the authors, in a limited number of experiments such as conducted in this paper, should compare Ad5, dl309, and some of their mutants.

Reviewer 3 Report

The authors have set out to examine the properties of a long established set of adenovirus 5 E1A mutants.The investigation appears to have been carefully carried out and has generated considerable amounts of data. Unfortunately, to my eyes at least, it tells us very little that is novel about the mode of action of AdE1A or adenovirus replication per se. Although I would not expect an in depth analysis of all the mutants a more detailed examination of at least one is required. Specifically, for example, the authors have shown that the deletion between CR1 and CR2 (dl1106) has pronounced effects on genome replication (Fig4) and yet was very inefficient at inducing S phase entry. A mass spec analysis of proteins binding to wt and dl1106 E1As could be very informative and would be straight forward to do. 

I also have a few specific criticisms which should be addressed:

1.A table outlining the properties of the mutant viruses obtained from previous studies should be included.

2. Figure 3 is unnecessary-the result is covererd adequately in the text.

3. The E1A expression in Figure 6 is difficult to understand-some attempt should be made to explain the multiple bands seen on the western blots. It seems that this is one of the most interesting observations in the manuscript, yet is hardly discussed (lines 269-71). As the authors note, the E1A protein levels do not reflect the RNA data shown in Figure 5 (which calls into question whether the latter is of value?). Considering the E1A western blots-do these results mean that the deletions affect the stability of the protein? I assume that the 2 major E1A bands seen at 16h with dl309 are the products of the 12S and 13S RNAs-so what is the very strong lower molecular weight band seen at 48h? Similarly, what is the origin of the 4 bands seen at 16h in dl1105 infection? The authors discuss E1A expression in their introduction but this is not really consistent with their observations; for example, the 12S and 13S products are considered to be expressed early in infection predominantly (ref9) but the major E1As in the dl309 sample are still visible at 72h.

4. The ability of the viruses to progress infected cells into S phase would be better examined by FACS analysis and this should be carried out. (If one were being pedantic analysis of RNA level does not necessarily tell you the protein expression of PCNA, BLM and MCM4 Figure 8).
